

**Hybrid improved EMD-BPNN model for the prediction of sea surface temperature**
Zhiyuan Wu [a,b,c], Changbo Jiang [a,c,*], Mack Conde [d], Bin Deng [a,c], Jie Chen [a,c]
a. School of Hydraulic Engineering, Changsha University of Science & Technology, Changsha, 410004, China;
b. School for Marine Science and Technology, University of Massachusetts Dartmouth, New Bedford, MA 02744, USA;
c. Key Laboratory of Water-Sediment Sciences and Water Disaster Prevention of Hunan Province, Changsha, 410004, China;
d. Department of Mathematics, University of Massachusetts Dartmouth, North Dartmouth, MA 02747, USA.
**Highlights**
• A novel SST predicting method based on the hybrid improved EMD algorithms and BP neural network
method are proposed in this paper.
• SST prediction results based on the hybrid EEMD-BPNN and CEEMD-BPNN models are compared and
discussed.
• Cases study of SST in the North Pacific shows that the proposed hybrid CEEMD-BPNN model can
effectively predict the time-series SST.
**Abstract:** Sea surface temperature (SST) is the major factor that affects the ocean-atmosphere interaction,
and in turn the accurate prediction of SST is the key to ocean dynamic prediction. In this paper, an SST
predicting method based on improved empirical mode decomposition (EMD) algorithms and back-
propagation neural network (BPNN) is proposed. Two different EMD algorithms have been applied
extensively for analyzing time-series SST data and some nonlinear stochastic signals. Ensemble empirical
mode decomposition (EEMD) algorithm and Complementary Ensemble Empirical Mode Decomposition
(CEEMD) algorithm are two improved algorithms of EMD, which can effectively handle the mode-mixing
problem and decompose the original data into more stationary signals with different frequencies. Each
Intrinsic Mode Function (IMF) has been taken as an input data to the back-propagation neural network model.
The final predicted SST data is obtained by aggregating the predicted data of individual IMF. A case study,
of the monthly mean sea surface temperature anomaly (SSTA) in the northeastern region of the North Pacific,
shows that the proposed hybrid CEEMD-BPNN model is much more accurate than the hybrid EEMD-BPNN
model, and the prediction accuracy based on BP neural network is improved by the CEEMD method.
Statistical analysis of the case study demonstrates that applying the proposed hybrid CEEMD-BPNN model



is effective for the SST prediction.

**Keywords.**
Sea Surface Temperature; Back-Propagation Neural Network; Empirical Mode Decomposition; Prediction;
Machine Learning Algorithms.

**1 Introduction**

The Sea Surface Temperature (SST) is a main factor in the interaction between the ocean and the

atmosphere (Wiedermann et al., 2017; He et al., 2017), and it characterizes the combined results of ocean
heat (Buckley et al., 2014; Griffies et al., 2015), dynamic processes (Takakura et al., 2018). It is a very
important parameter for climate change and ocean dynamics process, reflects sea-air heat and water vapor
exchange. Small changes in sea temperature can have a huge impact on the global climate. The well-known
El Niño and La Niña phenomena are caused by abnormal changes in SST (Chen et al., 2016a; Zheng et al.,

2016).

Therefore, scholars have begun to observe the SST in recent years, the observation of the SST is

important (Kumar et al., 2017; Sukresno et al., 2018). Accurate observation and effective prediction of the
SST are very important (Hudson et al., 2010). Predicting the SST in advance can enable people to take
appropriate measures to reduce the impact on daily life and reduce unnecessary losses. However, due to the
high randomness of the monthly mean sea surface temperature anomaly (SSTA), the nonlinear and non-
stationary characteristics are obvious. At present, there is no clear and feasible method with high accuracy to
effectively predict the SST (Zhu et al., 2015; Chen et al., 2016b; Khan et al., 2017).

In mathematics and science, a nonlinear system is a system in which the change of the output is not

proportional to the change of the input. Nonlinear dynamical systems, describing changes in variables over
time, may appear chaotic, unpredictable, or counterintuitive, contrasting with much simpler linear systems.
A stationary process is a stochastic process whose unconditional joint probability distribution does not change
when shifted in time. Consequently, parameters such as mean and variance also do not change over time. The
variation of SST is a deterministic non-linear dynamic system and a non-stationary time series data. Empirical
Mode Decomposition (EMD) is a state-of-the-art signal processing method proposed by Huang et al. (1998).
This method can decompose the signal data of different frequencies step by step according to the
characteristics of the data and obtain several periodic and trending signals orthogonal to each other, the

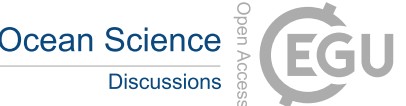

method can decompose the stronger nonlinear and non-stationary signals into weaker nonlinear and non-
stationary signals (Wang et al., 2015; Amezquita-Sanchez and Adeli,2015; Wang et al., 2016; Kim and Cho,
2016). However, there were some problems of the EMD method, such as mode mixing (Huang and Wu, 2008;
Wu and Huang, 2009).
To solve this problem, Wu and Huang (2009) proposed the Ensemble Empirical Mode Decomposition
(EEMD) method by adding different white noise in each ensemble member to suppress mode mixing. Yeh et
al. (2010) added two opposite-signal white noises to the time-series data sequence, and proposed an improved
algorithm for EEMD, Complete Ensemble Empirical Mode Decomposition (CEEMD). The decomposition
effect is equivalent to EEMD, and the reconstruction error caused by adding white noise is reduced (Tang et
al., 2015). At present, the EMD model and its improved algorithms had been widely used in many fields on
ocean science, such as storm surge and sea level rise (Wu et al., 2011; Lee, 2013; Ezer and Atkinson, 2014),
tidal amplitude (Cheng et al., 2017; Pan et al., 2018) and wave height (Duan et al., 2016; Sadeghifar et al.,
2017; López et al., 2017). These studies and applications reflected that the EMD model and its improved
algorithms can effectively reduce the non-stationarity of the time-series data, which helps further analysis
and processing.
For nonlinear prediction, the more commonly used methods are curve fitting (Motulsky and Ransnas,
1987), gray-box model (Pearson and Pottmann, 2000), homogenization function model (Monteiro et al.,
2008), neural network (Deo et al., 2001; Wang et al 2015; Kim et al., 2016) and so on. Among them, Back-
Propagation Neural Network (BPNN) (Lee, 2004; Jain and Deo, 2006; Savitha and Al, 2017; Wang et al.,
2018) has certain advantages in dealing with nonlinear problems, it is a basic machine learning algorithm
and its principle is simple and operability is strong, so in ocean science and engineering it has been widely
used.
In view of non-stationary and nonlinear monthly mean SST, the EEMD, CEEMD and BP neural network
will be used here to study how to improve the accuracy of SST prediction. The improved hybrid EMD-BPNN
models will be established for the prediction of SSTA in the northeastern region of the Pacific Ocean.

**2 Data collection**
The SST time-series data in this study is from NOAA Optimum Interpolation Sea Surface Temperature
(OISST) official website (Reynolds et al., 2007; Banzon et al., 2016; https://www.ncdc.noaa.gov/oisst/data-
access). The NOAA 1/4°daily OISST is an analysis constructed by combining observations from different




platforms (satellites, ships, buoys) on a regular global grid. There are two kinds of OISST, named after the
relevant satellite SST sensors. These are the Advanced Very High Resolution Radiometer (AVHRR) and
Advanced Microwave Scanning Radiometer on the Earth Observing System (AMSR-E); the AVHRR dataset
is used in this study. The average annual sea surface temperature in North Pacific (0°N-60°N, 100°E-100°W)
during January 1982 to December 2016 is shown in Fig.1.

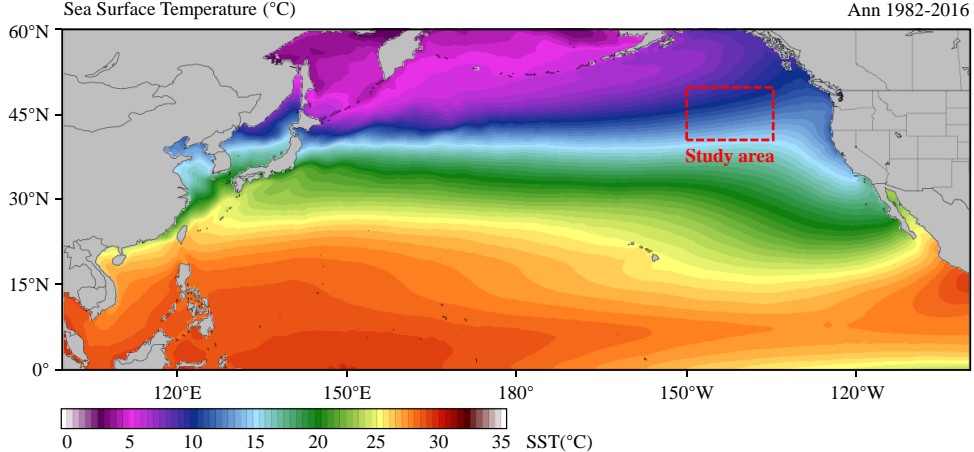

**Fig.1** Average annual sea surface temperature in North Pacific during Jan 1982 to Dec 2016 (35-years).

It has been shown that the sea surface temperature anomaly in the northeastern Pacific in the ten years

2006-2016 was 2.0°C warmer than in previous ten years 1996-2006. Previous studies (Bond et al., 2015)
showed that in the spring and summer of 2014, the high SST area of the northeastern Pacific had expanded
to coastal ocean waters, which affected the weather in coastal areas and the lives of fishermen, and even
affected the temperature in Washington, USA, causing interference to daily life.

In this study, we select the northeastern region of the North Pacific Ocean (in Fig.1, 40°N-50°N, 150°W-

135°W) to measure sea surface temperature. The time-series data of SST for the study area from January
1982 to December 2016 with a data length of 420 months was obtained from OISST-V2 (Fig. 2). The monthly
mean sea surface temperature anomaly (SSTA) was used in the analysis and calculation. As shown in Fig.
2(a), it can be found the overall time-series data is very messy, nonlinear and random from the perspective
of the image.



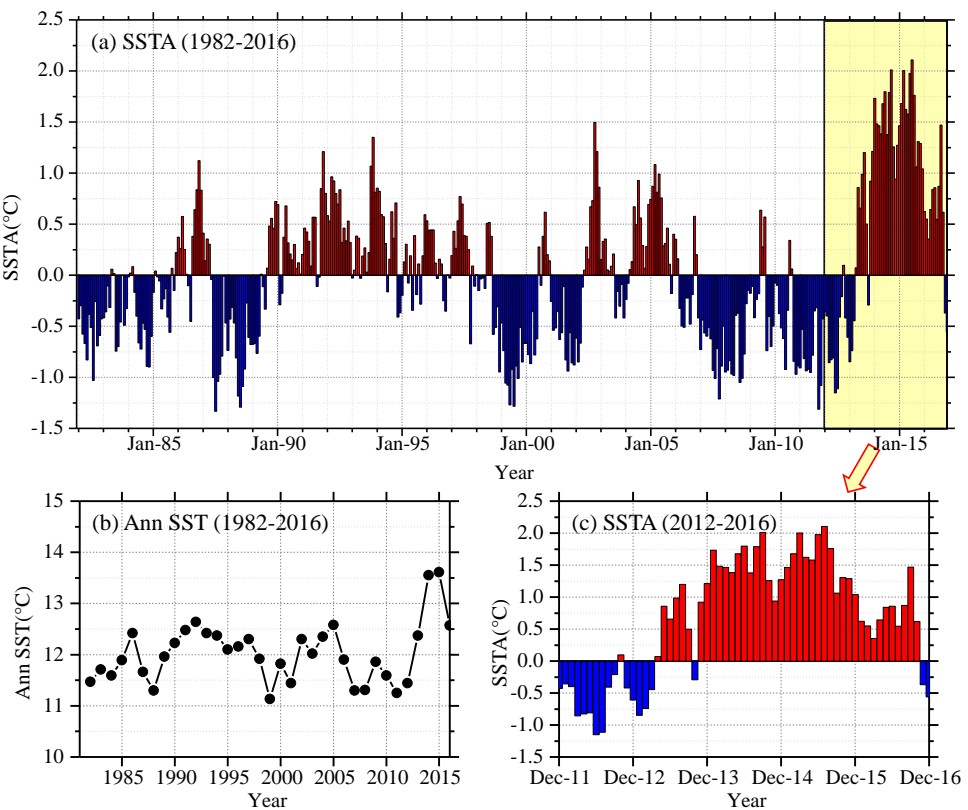

**Fig.2** The time-series of sea surface temperature in the study area. (a) SST anomaly (1982-2016, 35 years); (b) Annual SST (1982-2016, 35 years); (c) SST anomaly (2012-2016, 5 years).

## 3 Decomposition of SSTA

The purpose of this study is to combine the EEMD algorithm and the CEEMD decomposition algorithm respectively with the BP neural network algorithm to establish a new prediction model, an improved hybrid EMD-BPNN model. The EEMD and CEEMD algorithms are performed on the monthly mean SSTA data to obtain a series of intrinsic mode functions (IMFi). Each IMFi is predicted by a BP neural network and then each IMFi is reconstructed to obtain the predicted value of SSTA.

### 3.1 Decomposition by the EEMD algorithm

The SSTA in Fig. 2(a) has been decomposed based on the ensemble empirical mode decomposition (EEMD algorithm), and seven IMF components and a residual component RES (Residue) are obtained as shown in Fig. 3. It can be seen from Fig. 3 that the first three intrinsic mode function components IMF1,

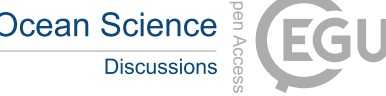



IMF2, and IMF3 still exhibit strong nonlinearity and non-stationarity. The IMF4 to IMF7 and the final trend
term RES have some periodicity and relatively regular fluctuation, and the non-stationary and nonlinear
properties are less than the first three components. The trend term RES reflects that the overall trend of SSTA
has gradually increased since 1982. As the non-stationarity of each IMFi is gradually reduced, the EEMD
algorithm will reduce the influence of non-stationarity on prediction. The absolute error (ERR) of the
decomposition can been calculated by the following Formula (1).
$$a(t) = \left| S(t) - \left[ \sum_{i=1}^{7} I_i(t) + R(t) \right] \right| \tag{1}$$

where, $a(t)$ is the absolute error (ERR), $S(t)$ the original SSTA observation data, $I_i(t)$ the $i$-th component

of the IMF (IMF$i$), and $R(t)$ the trend term (RES).

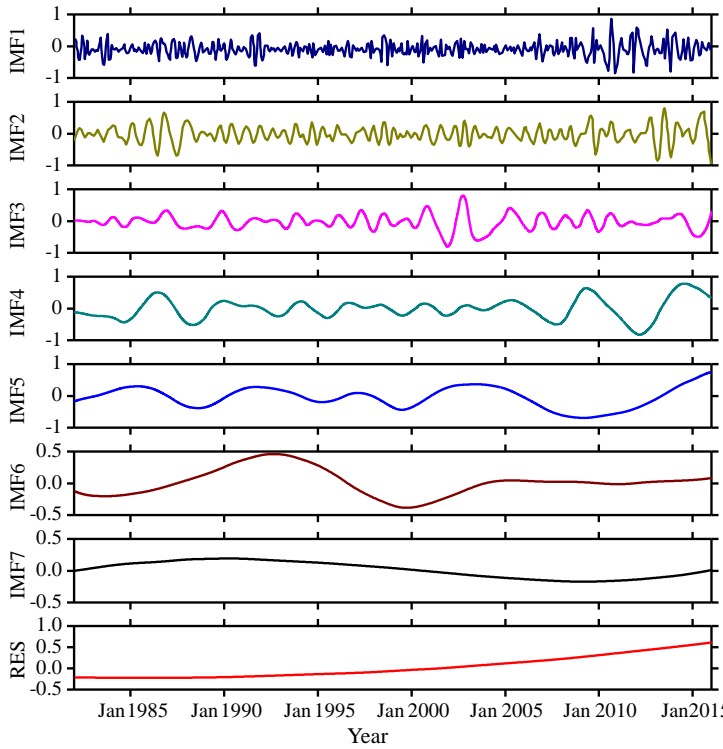


**Fig.3** IMF components and the trend item RES of monthly mean SSTA over the study area based on the
EEMD algorithm during 1982-2016.

The absolute error (ERR) based on EEMD algorithm is shown in Fig. 4. It can be seen from the figure





that the ERR of 420 months after decomposition is basically below 0.01 °C, and the ERR exceeds 0.01 °C in
five months: June 1989, September 1993, July 1998, May 1999 and March 2010.

In addition to June 1989, the other four monthly data with a large ERR occurred during the El Niño

period. The maximum error is in March 2010, the actual value is -0.1204 °C, the result based on EEMD
algorithm is -0.1325 °C, the ERR of decomposition is 0.0121 °C; the minimum error, in April 1987, is
$1.73 \times 10^{-5}$ °C. The overall mean ERR based on EEMD algorithm is 0.0035 °C and the order of magnitude is
$10^{-3}$.

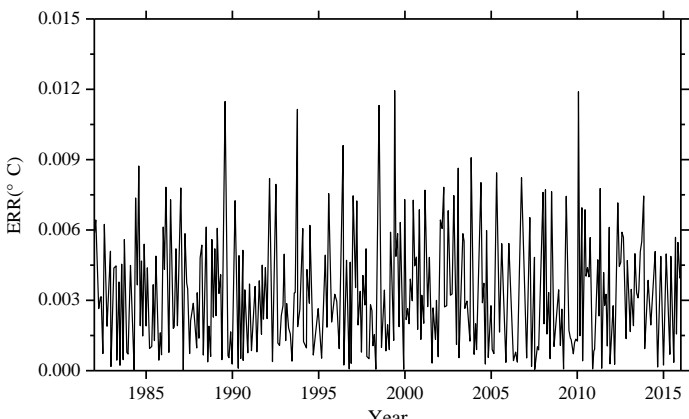


**Fig. 4** The ERR of monthly mean SSTA over the study area based on the EEMD algorithm during 1982-2016.

**3.2 Decomposition by the CEEMD algorithm**

The SSTA has been decomposed based on the complementary ensemble empirical mode decomposition

(CEEMD algorithm) and seven IMF components and a residual component RES (Residue) are obtained as
shown in Fig. 5. It can be seen when comparing the decomposition results based on EEMD and CEEMD
algorithms that although the mode components decomposed by CEEMD algorithm are different from the
corresponding results decomposed by EEMD, the nonlinearities and non-stationarities of the eight modes
decomposed by the two decomposition algorithms are gradually decreasing, and the final trend term RES is
an upward trend. Both decomposition algorithms confirm the characteristic of gradual increase for the overall
trend of the data series.

The absolute error (ERR) obtained based on the CEEMD algorithm is shown in Fig. 6. It can be seen

from the figure that the ERR of 420 months data after decomposition is less than $5 \times 10^{-16}$ °C, and the accuracy





is very better. The maximum error is $4.48\times10^{-16}$ °C in March 2016; the minimum error is zero. The overall
mean ERR based on CEEMD algorithm is $6.10\times10^{-17}$ °C and the order of magnitude is $10^{-17}$. By comparing
the results and errors of the above two decomposition algorithms, it can be seen that the error based on the
improved algorithm (CEEMD) is much smaller than the error based on EEMD algorithm. Because more
white noise with opposite sign had been added in CEEMD algorithm, the reconstruction error caused by
the white noise has been reduced over it in EEMD algorithm.

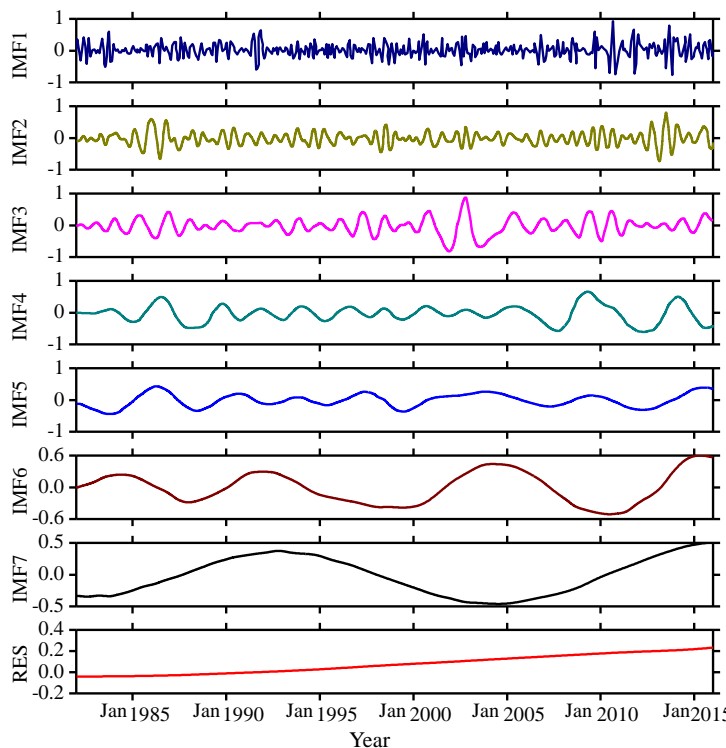


**Fig.5** IMF components and the trend item RES of monthly mean SSTA over the study area based on the
CEEMD algorithm during 1982-2016.





**Fig. 6** The ERR of monthly mean SSTA over the study area based on the CEEMD algorithm during 1982-2016.

## 4 SSTA prediction model

### 4.1 The BP neural network

Artificial Neural Network (ANN) is an information processing approach based on the biological neural network (López et al., 2015; Kim et al., 2016). In theory, ANN can simulate any complex nonlinear relationship through nonlinear units (neurons) and has been widely used in the prediction area, such as wave height and storm surge. The most basic structure of ANN consists of input layers, hidden layers and output layers. One of the most widely used ANN models is the back propagation neural network (BPNN, Wang et al., 2018) algorithm based on the BP algorithm.

The BPNN algorithm is a multi-layer feedforward network trained according to the error back propagation algorithm and is one of the most widely used deep learning algorithms. The BP network can be used to learn and store a large number of mappings of input and output models without the need to publicly describe the mathematical equations of these mapping relationships. The learning rule is to use the steepest descent method. When applied to SST predicting, the input data are monthly mean SST in previous months and the output data are predicted SST time-series data. The desired data for comparison is the observed actual SST.

### 4.2 SSTA prediction model based on hybrid improved EMD-BPNN algorithm

The proposed monthly mean sea surface temperature anomaly (SSTA) predicting model includes three



steps as follows. First, original SST datasets are decomposed into certain more stationary signals with
different frequencies by EEMD. Second, BP neural network is used to predict each IMF and the residue RES.
A rolling forecasting process is studied. The prediction is made using the previous data for one step ahead.
Finally, the prediction results of each IMF and the residue RES are aggregated to obtain the final SST
prediction results. The flowchart of SST prediction model based on hybrid improved empirical mode
decomposition algorithm (improved EMD algorithm) and back-propagation neural network (BPNN)is shown
in Fig. 7. The SST prediction model has been abbreviated as hybrid improved EMD-BPNN model in the
following article.

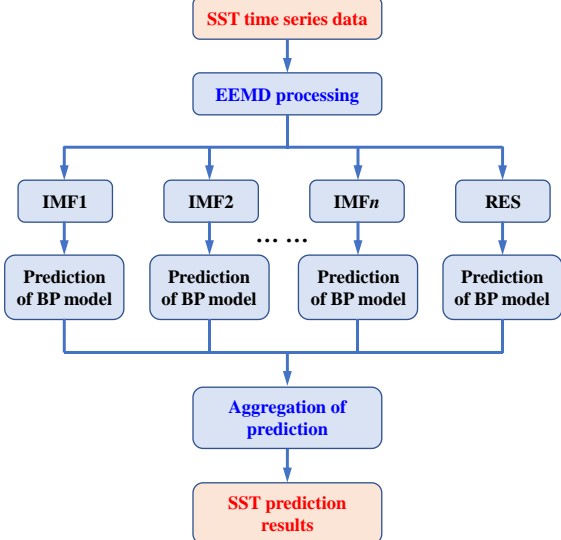


**Fig.7** The flowchart of SST prediction model based on hybrid improved empirical mode decomposition
algorithm (improved EMD algorithm) and back-propagation neural network (BPNN).

**5 Case study: SSTA prediction based on the hybrid improved EMD-BPNN models**
In order to study the effects of the two improved EMD algorithms (EEMD and CEEMD) on the
prediction results, and to analyze the prediction ability of BP neural network, the following experiments were
carried out. Predict SSTA results in 2017 and analyze the prediction abilities of different mode decomposition
data based on EEMD and CEEMD algorithms. The experiment content is as follows: the BP neural network
is trained with the decomposition data of each mode from 1982 to 2016, and the SSTA in 2017 is predicted
by the trained neural network, and the observation results of 12 months in 2017 is used to compare and



analyze with the prediction results.
Since the nonlinearity of the IMF1 to IMF3 is still relatively strong, a three-layer BP neural network
structure has been chosen and independently analyze and predict each month. For the IMF4 and subsequent
modes, since the nonlinearity and non-stationarity have been degraded relative to the first three modes, a BP
neural network with 12 nodes at input layer and output layer has been used to train and predict SSTA.
The prediction results of each mode decomposition component based on the EEMD algorithm are shown
in Fig. 8. The absolute errors of the predicted value and the actual value are shown in Table 1. Root mean
square error (RMSE) is used as metrics to access the performance of the two different models.

$$\text{RMSE} = \sqrt{\frac{1}{N}\sum_{n=1}^{N}\left(x_n - y_n\right)^2} \qquad (2)$$

where, $x_n$ and $y_n$ are the observed and the predicted values respectively, $N$ is the number of data used for
the performance evaluation. Results are shown in Table 1.

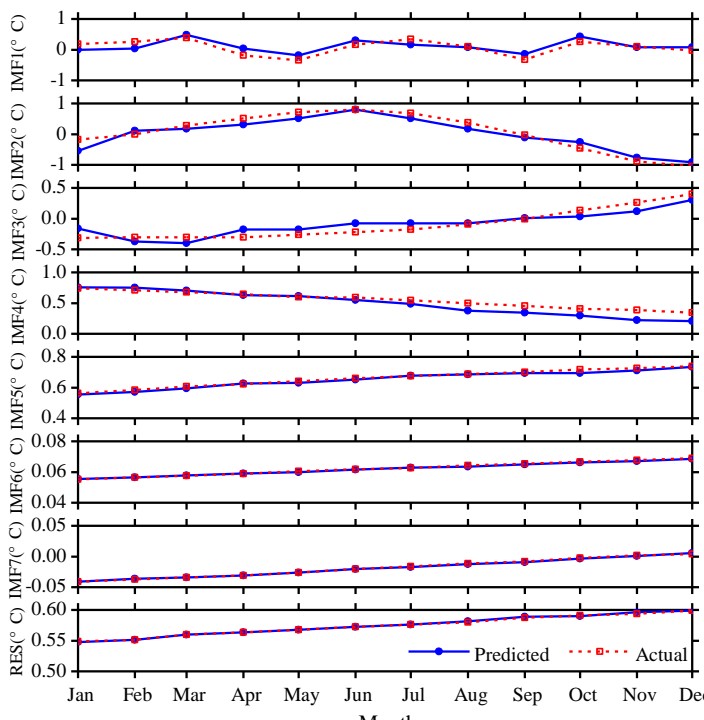

**Fig. 8** SSTA prediction results based on the hybrid EEMD-BPNN model of each individual component in

2017.




**Table 1.** The absolute errors ERRs of the SSTA prediction results of each individual component based on the
hybrid EEMD-BPNN model (unit: °C).

|  | Max ERR | Min ERR | Mean ERR | RMSE |
|---|---|---|---|---|
| IMF1 | 0.2197 | 0.0014 | 0.1424 | 0.1486 |
| IMF2 | 0.2166 | 0.0323 | 0.1297 | 0.1673 |
| IMF3 | 0.1872 | 0.0051 | 0.1070 | 0.1245 |
| IMF4 | 0.1602 | $1.6869 \times 10^{-4}$ | 0.0663 | 0.0857 |
| IMF5 | 0.0158 | 0.0010 | 0.0089 | 0.0104 |
| IMF6 | $3.8766 \times 10^{-4}$ | $1.9752 \times 10^{-4}$ | $2.7221 \times 10^{-4}$ | 0.0003 |
| IMF7 | $5.2662 \times 10^{-4}$ | $1.6387 \times 10^{-4}$ | $1.7907 \times 10^{-4}$ | 0.0002 |
| RES | $5.4859 \times 10^{-4}$ | $2.2308 \times 10^{-4}$ | $2.4766 \times 10^{-4}$ | 0.0002 |


It can be seen from Fig. 8 and Table 1 that the maximum absolute error (Max ERR) of the first
decomposition component IMF1 based on the hybrid EEMD-BPNN model is 0.2197 °C in January. The
minimum absolute error (Min ERR) is 0.0014 °C, which is in August. The prediction ability of the second
mode decomposition component IMF2 is roughly equivalent to the IMF1, and the mean absolute error (Mean
ERR) of the first three intrinsic mode function components IMF1, IMF2, and IMF3 are between 0.10 °C and
0.15 °C. The mean absolute errors of the IMF4 and IMF5 are 0.0663 °C and 0.0089 °C, respectively, and the
prediction accuracy based on the hybrid EEMD-BPNN model is roughly equivalent to the decomposition
accuracy of the EEMD algorithm. The prediction errors of the last two intrinsic mode function components
and the residue RES are on the order of $10^{-4}$. It can be seen that as the nonlinearity and non-stationarity of
the series data decreases, the error of the prediction results becomes smaller and smaller.
According to the same method, the eight mode components decomposed by CEEMD algorithm have
been analyzed and predicted. The prediction results and error analysis have been shown in Fig. 9 and Table
2. It can be seen from Fig. 9 and Table 2 that the maximum error of the first decomposition component IMF1
based on the hybrid CEEMD-BPNN model is 0.1779 °C in May. The minimum error is 0.0068 °C, which is
in June.
The prediction ability of the second mode decomposition component IMF2 is roughly equivalent to the
IMF1. Except for the four months of May, September, October, and November, the accuracies of prediction
results of other months are satisfactory. The prediction results of the first three intrinsic mode function



components IMF1, IMF2, and IMF3 are basically the same as the actual data. In the prediction results of the
fourth mode component IMF4, except for slight error in December, the prediction ability is better. The
predicted results of the last three intrinsic mode function components IMF5, IMF6, IMF7 and the residue
RES are basically consistent with the observation results.

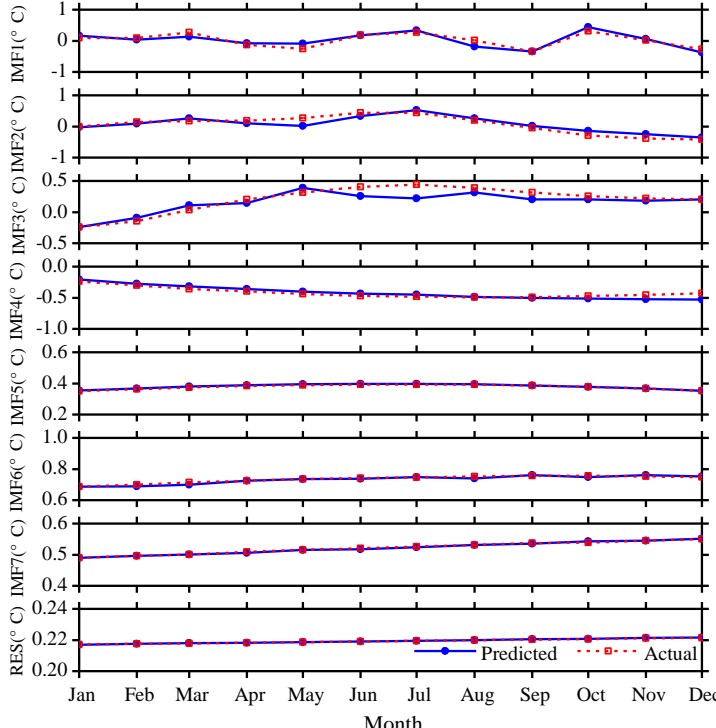


**Fig. 9** SSTA prediction results based on the hybrid CEEMD-BPNN model of each individual component in

2017.






**Table 2.** The absolute errors ERRs of the SSTA prediction results of each individual component based on the
hybrid CEEMD-BPNN model (unit: °C).

|      | Max ERR | Min ERR | Mean ERR | RMSE |
|------|---------|---------|----------|------|
| IMF1 | 0.1779 | 0.0068 | 0.0827 | 0.0987 |
| IMF2 | 0.1643 | 0.0413 | 0.0811 | 0.1124 |
| IMF3 | 0.1521 | 0.0160 | 0.0713 | 0.1006 |
| IMF4 | 0.0851 | 0.0211 | 0.0324 | 0.0427 |
| IMF5 | 0.0052 | $8.7694\times10^{-5}$ | 0.0021 | 0.0029 |
| IMF6 | 0.0103 | $5.7748\times10^{-5}$ | 0.0043 | 0.0056 |
| IMF7 | 0.0017 | $3.6026\times10^{-5}$ | $9.1374\times10^{-4}$ | 0.0010 |
| RES  | $3.0342\times10^{-5}$ | $2.0163\times10^{-6}$ | $1.1572\times10^{-5}$ | $1.5017\times10^{-5}$ |


The prediction results of the monthly mean SSTA in 2017 are obtained by reconstructing the mode

decomposition components (Fig. 10) and the absolute error (ERR) of prediction results has been shown in
Table 3. It can be seen from the figure and table that the prediction results based on the EEMD-BPNN model
have larger ERRs in January and August, exceeding 0.3 °C, and the accuracies of prediction results in other
months are satisfactory (the ERR is less than 0.3). The prediction accuracy based on the CEEMD-BPNN
model is satisfactory, except for the ERR exceeding 0.1 °C in October, and the prediction ability based on
the CEEMD-BPNN model is generally better than that of the EEMD-BPNN model.

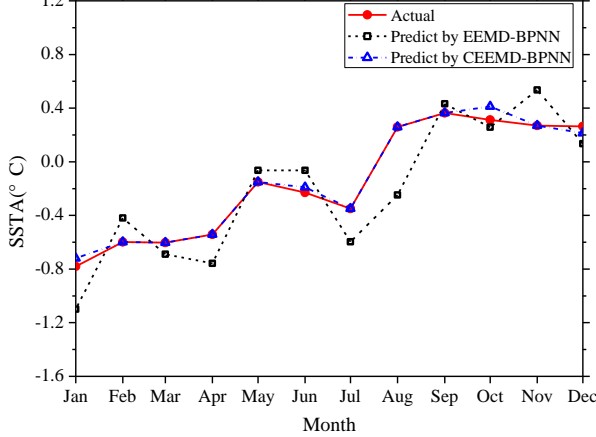


**Fig. 10** Monthly SSTA prediction results based on the hybrid improved EMD-BPNN models in 2017.




**Table 3.** The absolute errors ERRs of the SSTA prediction results based on the two different hybrid improved

EMD-BPNN models (unit: °C).

| | EEMD-BPNN model | CEEMD-BPNN model | | EEMD-BPNN model | CEEMD-BPNN model |
|---|---|---|---|---|---|
| Jan | 0.3188 | 0.0623 | Sep | 0.0687 | 0.0132 |
| Feb | 0.1780 | 0.0103 | Oct | 0.0545 | 0.1607 |
| Mar | 0.0867 | 0.0063 | Nov | 0.2651 | 0.0101 |
| Apr | 0.2153 | 0.0137 | Dec | 0.1290 | 0.0183 |
| May | 0.0854 | 0.0102 | **Min ERR** | **0.0545** | **0.0063** |
| Jun | 0.1662 | 0.0224 | **Max ERR** | **0.5068** | **0.1607** |
| Jul | 0.2474 | 0.0077 | **Mean ERR** | **0.1935** | **0.0289** |
| Aug | 0.5068 | 0.0112 | **RMSE** | **0.2299** | **0.0512** |


Correlation coefficient between the prediction values based on the CEEMD-BPNN model and

observations is shown that the value of the correlation coefficient that indicates a significance level of 0.001

and the correlation coefficient reached 0.97. The result   indicates that SSTA in 2017 had been predicted

accurately by the CEEMD-BPNN model. As can be seen from the above discussions, the ERR of

decomposition components based on the EEMD and CEEMD algorithms will affect the accuracy of the final

prediction results. Table 3 shows that predicting results of the hybrid CEEMD and BPNN model are

ameliorated a lot as compared to the EEMD-BPNN direct predicting model. This is because after CEEMD,

the original unsteady and nonlinear data are changed into certain components that have fixed frequency and

periodicity. The CEEMD algorithm with less decomposition error has less error in the final prediction results,

which proves that the CEEMD method has more advantages in data decomposition than the EEMD method.

At the same time, we can find that the final prediction error of the two prediction models mainly comes from

the first three mode decomposition components, and the error of the last five components has little effect on

the accuracy of the final prediction results.


**6 Conclusions**

This paper presents a novel SST predicting method based on the hybrid improved EMD algorithms and

BP neural network method to process the SST data with strong nonlinearity and non-stationarity. Through

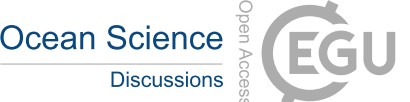



EEMD and CEEMD algorithms, SSTA time-series data are decomposed into different IMFs and a residue
RES. BP neural network is applied to predict individual IMFs and the residue RES. Final results can be
obtained by adding the predicting results of individual IMFs and RES.

In order to illustrate the effectiveness of the proposed approach, a case study was carried out. SSTA

prediction results based on the hybrid EEMD-BPNN model and hybrid CEEMD-BPNN model are discussed
respectively. In comparison, the proposed hybrid CEEMD-BPNN model is much better and its prediction
results are more accurate.

From the absolute error of the prediction results of each component IMF and the absolute error of the

predicted SSTA, the prediction error of SSTA mainly comes from the prediction of the first three mode
decomposition component (IMF1, IMF2 and IMF3), because the first three mode components still have
strong nonlinearity and non-stationarity. As the nonlinearity gradually decreases, the absolute error of the
prediction results gradually decreases.

SST prediction has been only preliminary carried out based on the two improved EMD algorithms and

BP neural network in this paper. The results show that the hybrid CEEMD-BPNN model is more accurate in
predicting SST. This work can provide a reference for predicting SST and El Niño in the future. In the follow-
up study, how to improve the forecast duration is the focus of this work.

**Acknowledgement**

This work was supported by National Natural Science Foundation of China (Grant Nos. 51809023,

51879015, 51839002, 51809021 and 51509023).

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
