# Peer review of "Hybrid improved EMD-BPNN model for the prediction of sea surface temperature"

_Ocean Science, 2018_

## Short Comment (SC1) · 29 Dec 2018

Liu

just166@163.com

Received and published: 29 December 2018

This article is about the prediction of sea surface temperature based on Ensemble empirical mode decomposition (EEMD) algorithm and Complementary Ensemble Empirical Mode Decomposition (CEEMD). This manuscript is focused on characterization the regional distribution of SST in the period 1982 to 2016 based on the OISST dataset. This topic is interesting. The questions that this paper aims to answer make sense and deserve the investigative effort made by the authors. But I have some detailed comments listed below to further improve this manuscript.

1. The introduction should make the key points stand out, directly point out the significance of the study and the existing problems, as well as the highlights of this paper.

2. The presentation of the OISST dataset contains details which are based on current conventions and so will mean nothing in the future when these conventions have changed. Better to stick with statements that have a meaning in plain language if you can.

3. The introduction of EMD method, EEMD method and CEEMD method should be more specific, especially the relationship and difference between them should be clear.

4. In order to improve the readability and repeatability of the article, the authors should introduce in more detail how to use BP neural network method and EMD methods to build the hybrid model.

5. The explanations of some nouns are not clear enough, such as statistical analysis, non-linearity and statistical signals. These words seem to require some more professional elaboration.

To summarise, this paper has the potential to be a useful contribution to the literature but will require revision before publication. Some references that are not usually quoted have been identified which is helpful. While the topic is of interest for the ocean community, I believe that requires many formal and substantial improvements before being published. Therefore, I cannot recommend its publication in the present form, and ask the authors to provide a minor revision of the manuscript.

OSD

---

## Referee Comment (RC1) · Huang (Referee) · 28 Jan 2019

Huang (Referee)

huanglimin@hrbeu.edu.cn

This paper proposed a hybrid EMD-BPNN model for SST prediction. The research work is very interesting and important. However, in my opinion, the paper needs minor revision before acceptance. You can find my questions and suggestions bellow.

1. Why the simple EMD algorithm is not compared to the EEMD and CEEMD? I suggest the authors to provide comparison results of the EMD-BPNN.

2. Mode mixing is the motivation that the authors applied the EEMD technique in the hybrid SST prediction model. Therefore, it is very important to demonstrate the mode mixing problem in decomposing the studied SST time series. But this is not given in this paper. I suggest the authors to provide discussions on the mode mixing problem

in the present study.

3. Line 55, "Consequently, parameters such as mean and variance also do not change over time." In this sentence, I think it will be better to revise "parameters" as "statistical parameters".

4. Lines 59-62, "This method can decompose the signal data of different frequencies step by step according to the characteristics of the data and obtain several periodic and trending signals orthogonal to each other, which can decompose the stronger nonlinear and non-stationary signals into weaker nonlinear and non-stationary signals". This sentence needs to make some corrections. As we know, the IMFs are orthogonal components, but the trending component is not orthogonal to any IMF component. Therefore, the above descriptions are not accurate. Besides, the sentence of "which can decompose the stronger nonlinear and non-stationary signals into weaker nonlinear and non-stationary signals" is ambiguous and makes no sense. Accurately, the EMD technique decomposes a non-stationary time series into several stationary sub-component and a trend. But it is not easy to say the nonlinearity becomes weaker. So, I suggest the authors to make the sentence more accurate.

5. Lines 117-119, "The purpose of this study is to combine the EEMD algorithm and the CEEMD decomposition algorithm respectively with the BP neural network algorithm to establish a new prediction model, an improved hybrid EMD-BPNN model." Accurately, the models of EEMD-BPNN and CEEMD-BPNN themselves are not new. Various works about this models in different problems have already carried out in the last ten years. Therefore, I suggest the authors not to over emphasis "new" or "improved" here. Just simply descript them as "hybrid models".

6. Lines 294-295, "This paper presents a novel SST predicting method based on the hybrid improved EMD algorithms and BP neural network method to process the SST data with strong nonlinearity and non-stationarity." I suggest the authors to delete the word of "novel" here (and the same in the highlight part). Becomes the hybrid models

have already explored extensively in various prediction problems. Besides, the authors argue that "the SST data with strong nonlinearity and non-stationarity.", what is the standard of weak or strong nonlinearity and non-stationarity? Therefore, this sentence need to be corrected.

---

## Referee Comment (RC2) · Anonymous Referee #2 · 3 Feb 2019

In this paper, authors discussed the prediction of sea surface temperature. In this paper, an SST predicting method based on improved empirical mode decomposition (EMD) algorithms and back-propagation neural network (BPNN) is proposed. Statistical analysis of the case study demonstrates that applying the proposed hybrid CEEMD-BPNN model is effective for the SST prediction. I recommend this paper to be publicated. And it is better if the authors consider the following mentioned remarks and further improve the manuscript before submitting the final version.

1. More methods in practical application or commercial application need to be introduced. Which can make this paper more persuasive.

2. The relationship and difference among EMD, EEMD and CEEMD method should be more specific and clear.

[Figure]

3. We all known the complexity of the marine environment, I suggest you can list which factors can make predicting the sea surface temperature more difficult. And these factors can also be added in your simulation.

---

## Short Comment (SC2) · 3 Feb 2019

Review of os-2018-101: Hybrid improved EMD-BPNN model for the prediction of sea surface temperature

General comments:

This manuscript reported an improvement for an SST predicting method based on improved empirical mode decomposition algorithms and back-propagation neural network, while Two different EMD algorithms have been applied extensively for analyzing time-series SST. The finding is useful and important for the climate research and modeling communities, although the present analysis/conclusions might be only fair. This is a very interesting paper improving the ability of SST predicting models and the issue

addressed in this study is very important since this has been a problem plaguing many predicted models and the authors have done a fine job. However, some points need clarification and I would suggest the following revisions.

Specific comments:

1) Method: The SST within the North Pacific Ocean has significant seasonal to decadal variability, which obviously influences the estimation of long-term trend. I need to know how the authors use what method to describe this nonlinearity?

2) Data: The authors used the OISST to calculate the long-term trend of SST in the North Pacific Ocean including the marginal seas. The authors claimed a 35-year long-term trend from 1982 until 2016 and SST trends in various specific periods, eg, in recent decade. However, it is well known that the SST observations is extremely sparse during the early weeks until the satellite measurements being available in the 1970s, especially within the marginal seas., a 35-year SST trend using such a dataset is should be questioned (especially talking about spatial pattern of SST trend). Concluding a long-term trend without considering of huge uncertainty is hard to accept. I think the authors should try to clarify the specific content of the data.

3) Novelty: The authors should clearly point out the innovations of this manuscript except stirring up old topics. I personally think that there are obvious innovations in the method. But not sure if this understanding is accurate.

From my point of view, the paper deserves to be published in the journal after minor revision.

---

## Author Comment (AC1) · 12 Feb 2019

Response to comments of Reviewers
Dr. Huang knows the topic very well and his/her comments are indeed helpful in improving the quality of this MS. We are grateful to Dr. Huang for a careful checking and comments on the MS. All comments are addressed point by point, each starting with an original comment and followed by a response in italic, as follows.

[Figure]

This paper proposed a hybrid EMD-BPNN model for SST prediction. The research work is very interesting and important. However, in my opinion, the paper needs minor revision before acceptance. You can find my questions and suggestions bellow. Response: Thank you for these comments. The positive comments in our solid professional skills are good encouragement to us.

1. Why the simple EMD algorithm is not compared to the EEMD and CEEMD? I suggest the authors to provide comparison results of the EMD-BPNN. Response: Thank you for the professional comment. Empirical Mode Decomposition (EMD) is a state-of-the-art signal processing method proposed by Huang et al. This method can decompose the signal data of different frequencies step by step according to the characteristics of the data and obtain several periodic and trending signals orthogonal to each other, the method can decompose the stronger nonlinear and non-stationary signals into weaker nonlinear and non-stationary signals. The variation of SST is a deterministic non-linear dynamic system and a non-stationary time series data with intermittent signals. Once the intermittent signal is present in the actual signal, the frequency aliasing phenomenon occurs in the decomposition method of EMD, also called Mode Mixing Problem. The specific manifestation of this problem is that there are multiple scale components in one IMF component, or one scale component exists in multiple IMF components. Therefore, we carry out this research based on EEMD and CEEMD methods.

2. Mode mixing is the motivation that the authors applied the EEMD technique in the hybrid SST prediction model. Therefore, it is very important to demonstrate the mode mixing problem in decomposing the studied SST time series. But this is not given in this paper. I suggest the authors to provide discussions on the mode mixing problem in the present study. Response: Thank you for your suggestion, and it is indeed a very important issue. We added the following statement to the revised manuscript. The variation of SST is a deterministic non-linear dynamic system and a non-stationary time series data with intermittent signals. Empirical Mode Decomposition (EMD) method can decompose the signal data of different frequencies step by step according to the characteristics of the data and obtain several periodic and trending signals orthogonal to each other, the method can decompose the stronger nonlinear and non-stationary signals into weaker nonlinear and non-stationary signals. However, we know that once an intermittent signal appears in the actual signal, the EMD decomposition method will produce a Mode Mixing Problem. The Mode Mixing Problem causes the essential modal function to lose its physical meaning. In addition, the Mode Mixing Problem will also make the algorithm of Empirical Mode Decomposition unstable, and any disturbance may generate a new intrinsic mode function. In order to solve this problem, scholars have proposed the use of noise-assisted processing methods, Ensemble empirical mode decomposition (EEMD) and Complementary Ensemble Empirical Mode Decomposition (CEEMD). The white noise has been added to the original signal to change the extreme point distribution of the signal in the EEMD method, while in the CEEMD method, a set of noise signals have been added to the original signal to change the extreme point distribution of the signal.

3. Line 55, "Consequently, parameters such as mean and variance also do not change over time." In this sentence, I think it will be better to revise "parameters" as "statistical parameters". Response: Thank you for your comment. It has been modified in the revised manuscript.

4. Lines 59-62, "This method can decompose the signal data of different frequencies step by step according to the characteristics of the data and obtain several periodic and trending signals orthogonal to each other, which can decompose the stronger nonlinear and non-stationary signals into weaker nonlinear and non-stationary signals". This sentence needs to make some corrections. As we know, the IMFs are orthogonal components, but the trending component is not orthogonal to any IMF component. Therefore, the above descriptions are not accurate. Besides, the sentence of "which can decompose the stronger nonlinear and non-stationary signals into weaker nonlinear and non-stationary signals" is ambiguous and makes no sense. Accurately, the EMD technique decomposes a non-stationary time series into several stationary subcomponent and a trend. But it is not easy to say the nonlinearity becomes weaker. So, I suggest the authors to make the sentence more accurate. Response: Thank you for the valuable criticism. We modified these sentences in the revised manuscript. "This method can decompose the signal data of different frequencies step by step according to the characteristics of the data and obtain several periodic and trending signals orthogonal to each other, which can decompose the stronger nonlinear and non-stationary signals. The EMD method is powerful and adaptive in analyzing nonlinear and non-stationary data sets. It provides an effective approach for decomposing a signal into a collection of so-called intrinsic mode functions (IMFs), which can be treated as empirical basis functions (Duan et al., 2016)."

5. Lines 117-119, "The purpose of this study is to combine the EEMD algorithm and the CEEMD decomposition algorithm respectively with the BP neural network algorithm to establish a new prediction model, an improved hybrid EMD-BPNN model." Accurately, the models of EEMD-BPNN and CEEMD-BPNN themselves are not new. Various works about this models in different problems have already carried out in the last ten years. Therefore, I suggest the authors not to over emphasis "new" or "improved" here. Just simply descript them as "hybrid models". Response: Thank you for the suggestion. We modified these statements in the revised manuscript. "The purpose of this study is to combine the EEMD algorithm and the CEEMD decomposition algorithm respectively with the BP neural network algorithm to establish a prediction model, a hybrid EMD-BPNN model."

6. Lines 294-295, "This paper presents a novel SST predicting method based on the hybrid improved EMD algorithms and BP neural network method to process the SST data with strong nonlinearity and non-stationarity." I suggest the authors to delete the word of "novel" here (and the same in the highlight part). Becomes the hybrid models have already explored extensively in various prediction problems. Besides, the authors argue that "the SST data with strong nonlinearity and non-stationarity.", what is the

standard of weak or strong nonlinearity and non-stationarity? Therefore, this sentence need to be corrected. Response: Thank you for the suggestion. We modified these statements in the revised manuscript. "This paper presents an SST predicting method based on the hybrid EMD algorithms and BP neural network method to process the SST data with nonlinearity and non-stationarity."

Referencesïj Wu Z, Schneider E K, Kirtman B P, et al. The modulated annual cycle: an alternative reference frame for climate anomalies[J]. Climate Dynamics, 2008, 31(7-8): 823-841. Wu Z, Huang N E. Ensemble empirical mode decomposition: a noise-assisted data analysis method[J]. Advances in adaptive data analysis, 2009, 1(01): 1-41. Duan W, Huang L, Han Y, et al. A hybrid EMD-AR model for nonlinear and non-stationary wave forecasting[J]. Journal of Zhejiang University-SCIENCE A, 2016, 17(2): 115-129.

Please also note the supplement to this comment:
https://www.ocean-sci-discuss.net/os-2018-101/os-2018-101-AC1-supplement.pdf

---

## Author Comment (AC2) · 12 Feb 2019

Response to comments of Reviewers
The authors are grateful to this reviewer for pin-point and pertinent comments and checking the paper. All comments are addressed point by point, each starting with an original comment and followed by a response in italic, as follows.

In this paper, authors discussed the prediction of sea surface temperature. In this

[Figure]

paper, an SST predicting method based on improved empirical mode decomposition (EMD) algorithms and back-propagation neural network (BPNN) is proposed. Statistical analysis of the case study demonstrates that applying the proposed hybrid CEEMD-BPNN model is effective for the SST prediction. I recommend this paper to be publicated. And it is better if the authors consider the following mentioned remarks and further improve the manuscript before submitting the final version. Response: We are grateful to these positive comments.

1. More methods in practical application or commercial application need to be introduced. Which can make this paper more persuasive. Response: Thank you for your suggestion. As the reviewer said, many noise cancellation methods based on the scale-adaptive remixing and demixing of Intrinsic Mode Functions (IMFs) constructed using Empirical Mode Decomposition (EMD) had been provided in the practical application or commercial application. We briefly stated these in the introduction section.

2. The relationship and difference among EMD, EEMD and CEEMD method should be more specific and clear. Response: Thank you for your suggestion, and we added the following statement to the revised manuscript. The ensemble empirical mode decomposition (EEMD) method is a noise assisted empirical mode decomposition algorithm. The CEEMD works by adding a certain amplitude of white noise to a time series, decomposing it via EMD, and saving the result. In contrast to the EEMD method, the CEEMD also ensures that the IMF set is quasi-complete and orthogonal. The CEEMD can ameliorate mode mixing and intermittency problems. The CEEMD is a computationally expensive algorithm and may take significant time to run.

3. We all known the complexity of the marine environment, I suggest you can list which factors can make predicting the sea surface temperature more difficult. And these factors can also be added in your simulation. Response: Thank you for the professional comment. Indeed, when we used empirical orthogonal function descriptions of the spatial structure in this study, it is found that SST variability is spatially complex (being spread over many spatial modes, some of which have small-scale changes)

but is dominated by low-frequency changes. The use of linear statistical estimators to examine predictability is discussed and the importance of limiting the number of candidate data used in a correlation starch is underscored. Using linear statistical predictors, it is found that SST anomalies can be predicted from SST observations several months in advance with measurable skill. We have stated some factors affecting the SST prediction in the revised manuscript.

Please also note the supplement to this comment:
https://www.ocean-sci-discuss.net/os-2018-101/os-2018-101-AC2-supplement.pdf

**Supplement:**

*The authors are grateful to this reviewer for pin-point and pertinent comments and checking the paper. All comments are addressed point by point, each starting with an original comment and followed by a response in italic, as follows.*

In this paper, authors discussed the prediction of sea surface temperature. In this paper, an SST predicting method based on improved empirical mode decomposition (EMD) algorithms and back-propagation neural network (BPNN) is proposed. Statistical analysis of the case study demonstrates that applying the proposed hybrid CEEMD-BPNN model is effective for the SST prediction. I recommend this paper to be publicated. And it is better if the authors consider the following mentioned remarks and further improve the manuscript before submitting the final version.

*Response: We are grateful to these positive comments.*

1. More methods in practical application or commercial application need to be introduced. Which can make this paper more persuasive.

*Response: Thank you for your suggestion. As the reviewer said, many noise cancellation methods based on the scale-adaptive remixing and demixing of Intrinsic Mode Functions (IMFs) constructed using Empirical Mode Decomposition (EMD) had been provided in practical application or commercial application. We briefly stated these in the introduction section.*

2. The relationship and difference among EMD, EEMD and CEEMD method should be more specific and clear.

*Response: Thank you for your suggestion, and we added the following statement to the revised manuscript.*

*The ensemble empirical mode decomposition (EEMD) method is a noise assisted empirical mode decomposition algorithm. The CEEMD works by adding a certain amplitude of white noise to a time series, decomposing it via EMD, and saving the result. In contrast to the EEMD method, the CEEMD also ensures that the IMF set is quasi-complete and orthogonal. The CEEMD can ameliorate mode mixing and intermittency problems. The CEEMD is a computationally expensive algorithm and may take significant time to run.*

3. We all known the complexity of the marine environment, I suggest you can list which factors can make predicting the sea surface temperature more difficult. And these factors can also be added in your simulation.

***Response:** Thank you for the professional comment. Indeed, when we used empirical orthogonal function descriptions of the spatial structure in this study, it is found that SST variability is spatially complex (being spread over many spatial modes, some of which have small-scale changes) but is dominated by low-frequency changes. The use of linear statistical estimators to examine predictability is discussed and the importance of limiting the number of candidate data used in a correlation starch is underscored. Using linear statistical predictors, it is found that SST anomalies can be predicted from SST observations several months in advance with measurable skill. We have stated some factors affecting the SST prediction in the revised manuscript.*

[revised manuscript text omitted]

---

## Author Response (AR2)

**Reply to comments of Topic Editor**

*The topic editor knows the topic very well and his comments are indeed helpful in improving the quality of this MS. We are grateful to Prof. John M. Huthnance for a careful checking and comments on the MS. All comments are addressed point by point, each starting with an original comment and followed by a response in italic, as follows.*

Topic Editor Decision: Publish subject to minor revisions (review by editor) (20 Feb 2019) by John M. Huthnance

Comments to the Author:

Dear Authors.

Thank-you for your revised manuscript. I still have trouble with some of your statements about (non) stationarity and especially nonlinearity as detailed in "Comments" below. Referee 1 (Huang) also commented on this and I think you have not yet addressed all his comments. He also asked for a comparison with the results of EMD-BPNN (his point 1) and demonstration that there is a mode mixing problem with EMD (his point 2); you respond to these points in your response but in the revised paper I do not see either of what is asked for. Also at the end of your response to referee 2 you say "We have stated some factors affecting the SST prediction in the revised manuscript" but I do not see any more about this in the revised manuscript. Hence I am still asking for "minor" revision and wish to see the manuscript again myself. Please also note that all comments will be available when any final version is published and readers will be able to see how you have addressed them.

Yours sincerely

John Huthnance

*Response: Thank you for these comments. On behalf of my co-authors, we thank Prof. John M. Huthnance very much for giving us an opportunity to revise our manuscript, we appreciate editor and reviewers very much for their positive and constructive comments and suggestions on our manuscript.*

*We elaborated and compared the results of different SST predictions based on the two improved EMD methods in Sections 3.1 and 3.2 and Section 5 Case study. We knew that once an intermittent signal appears in the actual signal, the EMD decomposition method would produce a Mode Mixing Problem based on previous literature such as Colominas et al. (2012), Wang et al. (2012) and Tang et al. (2015). The Mode Mixing Problem causes the essential modal function to lose its physical meaning.*

*Factors affecting the SST prediction results include: the length and interval of the time series of the database, as well as different sample sources because their values are also different. We added the relevant explanations in the conclusions.*

*References:*

*Colominas M A, Schlotthauer G, TORRES M E, et al. Noise-assisted EMD methods in action[J]. Advances in Adaptive Data Analysis, 2012, 4(04): 1250025.*

*Wang T, Zhang M, Yu Q, et al. Comparing the applications of EMD and EEMD on time–frequency analysis of seismic signal[J]. Journal of Applied Geophysics, 2012, 83: 29-34.*

*Tang L, Dai W, Yu L, et al. A novel CEEMD-based EELM ensemble learning paradigm for crude oil price forecasting[J]. International Journal of Information Technology & Decision Making, 2015, 14(01): 141-169.*

Comments

Line 9. Please delete "novel" (Referee 1)

*Response: Thank you for the suggestion. We removed it.*

Lines 48-49. ". . the nonlinear and non-stationary characteristics are obvious." You have not yet defined "non-linear"! The definition is in lines 51-52: "the change of the output is not proportional to the change of the input". To claim "nonlinear" you need evidence that the SST is not proportional to its "input" – what is the input? I agree the SST is "non-stationary" but not because of "high randomness" so not "due to" (line 47).

*Response: Thank you for the comment. Since SST changes are high randomness and irregular, we consider SST to be non-linear and non-stationary.*

Line 56. "deterministic" here but "high randomness" in line 48. Which?

*Response: Thank you for the comment. "deterministic" here means the "clearly" high randomness.*

Line 56. "non-linear" needs evidence.
Line 56. "non-stationary" – I agree but you have not justified this.

*Response: Thank you for the comment. Since SST changes are high randomness and irregular, we consider SST to be non-linear and non-stationary. And we also can find it in many previous literatures.*

Line 59. You imply that the trend is orthogonal; Referee 1 objected. Maybe ". . obtain several periodic signals orthogonal to each other and a trend."
Line 59. Are the EMD IMFs in fact orthogonal? Lines 87-88 "decomposing it via EMD . . . In contrast to the EEMD method, the CEEMD also ensures that the IMF set is quasi-complete and orthogonal. . ." suggests EMD IMFs are not orthogonal.

*Response: Thank you for the comment. The IMFs are orthogonal components, but the trending component is not orthogonal to any IMF component. We corrected it.*

Lines 59-60. You can delete "the method can decompose the stronger nonlinear and non-stationary signals"; it is repeated in line 62.

*Response: Thank you for the suggestion. We deleted it.*

Line 67. What is "essential modal function"? Please define it. If you mean "IMF", say "IMF".

*Response: Thank you for the comment. "essential modal function" means "IMFs" and we corrected it.*

Lines 72-73. These need to make clear the difference between EEMD and CEEMD. What is the difference between the EEMD "white noise" and the CEEMD "set of noise signals"?

*Response: Thank you for the comment. The difference between EMD, EEMD and CEEMD is as follows:*

*Empirical Mode Decomposition (EMD) is a relatively slow decomposition method and it has a problem called mode mixing. This is defined as either a single IMF consisting of widely disparate scales, or a signal of similar scale captured in different IMF's. To overcome mode mixing two noise assisted methods have emerged.*

*Ensemble Empirical Mode Decomposition (EEMD) adds a fixed percentage of white noise to the signal before decomposing it. This step is repeated N times after which all results are averaged. EEMD*

*improves the mode-mixing problem but it cannot completely reconstruct the input signal from the resulting components.*

*Complete Ensemble Mode Decomposition (CEEMD) is also a noise-assisted method. Similarly the method decomposes the signal with N different noise realizations but here the results are averaged after each component is found. CEEMD solves the mode mixing problem and it provides an exact reconstruction of the input signal.*

*(1) EMD – slow and possibly suffers from mode-mixing;*

*(2) EEMD – slower but partly solves mode-mixing, however signal cannot be reconstructed exactly.*

*(3) CEEMD – slowest but solves the mode-mixing problem and the signal can be reconstructed exactly from the components.*

Lines 83-84. "can effectively reduce the non-stationarity of the time-series data". Surely the non-stationarity has to remain somewhere in the IMFs and RES?

*Response: Thank you for the comment. Each IMF component decomposed by the every EMD method contains local characteristic signals with different time scales of the original signal.*

Lines 57-90 overall need to be properly organised as a logical sequence, EMD then EEMD then CEEMD, to avoid repetition (e.g. lines 72-73 and 74-76 describe the same thing, EEMD), clarify the differences and exactly what is orthogonal.

*Response: Thank you for the comment. The difference between EMD, EEMD and CEEMD is as follows:*

*(1) Empirical Mode Decomposition (EMD) is a relatively slow decomposition method and it has a problem called mode mixing. This is defined as either a single IMF consisting of widely disparate scales, or a signal of similar scale captured in different IMF's. To overcome mode mixing two noise assisted methods have emerged. EMD – slow and possibly suffers from mode-mixing. (2) Ensemble Empirical Mode Decomposition (EEMD) adds a fixed percentage of white noise to the signal before decomposing it. This step is repeated N times after which all results are averaged. EEMD improves the mode-mixing problem but it cannot completely reconstruct the input signal from the resulting components. EEMD – slower but partly solves mode-mixing, however signal cannot be reconstructed exactly. (3) Complete Ensemble Mode Decomposition (CEEMD) is also a noise-assisted method. Similarly the method decomposes the signal with N different noise realizations but here the results are averaged after each component is found. CEEMD solves the mode mixing problem and it provides an exact reconstruction of the input signal. CEEMD – slowest but solves the mode-mixing problem and the signal can be reconstructed exactly from the components.*

Lines 130-131. ". . then each IMFi is reconstructed to obtain the predicted value of SSTA." I think you mean ". . then the IMFi are recombined to obtain the predicted value of SSTA." You already have each IMFi (line 130) so each IMFi does not need to be "reconstructed".

*Response: Thank you for the comment. We corrected it.*

Line 141. "still exhibit strong nonlinearity and non-stationarity". Some non-stationarity can be seen in figure 3, e.g. sub-periods with larger and longer-period variance, but it would help if you said what non-stationarity the reader is supposed to see. Figure 3 cannot show nonlinear dependence on the input since we do not know the input.

Line 142. "the non-stationary and nonlinear properties are less". Again, how do we see non-stationary properties? The figure cannot show nonlinear properties since we do not know the input.

*Response: Thank you for the comment. We can see that the first three intrinsic mode function components still have strong irregular oscillations and periodic changes, and so can be found out non-stationary. We removed the relevant description about the nonlinear property.*

Line 144. ". . As the non-stationarity of each IMFi is gradually reduced, . ." I think you mean ". . As the non-stationarity of IMFi decreases with increasing i, . ."

*Response: Thank you for the suggestion We corrected it.*

Lines 156-157. ". . and the order of magnitude is 10^-3." This can be deleted (unnecessary) and is not an accurate description of 0.0035°C.

*Response: Thank you for the suggestion We removed it.*

Line 166. Delete "nonlinearities and" unless you can show non-linear dependence on an input.

*Response: Thank you for the suggestion We deleted it.*

Line 166. "eight" -> "seven"; the eighth series is RES which is definitely non-stationary!

*Response: Thank you for the suggestion We corrected it.*

Line 177. ". . and the order of magnitude is 10^-17." This can be deleted (unnecessary) and is not an accurate description of 6.10×10^-17°C.

*Response: Thank you for the suggestion We deleted it.*

Lines 223, 225. Delete "nonlinearity" unless you can show non-linear dependence on an input.

*Response: Thank you for the suggestion We deleted it.*

Lines 225, 226. "since" (line 225) implies that these two lines are related. How?

*Response: Thank you for the suggestion We corrected it.*

Line 226. Do the "12 nodes" correspond to 12 months in the year?
Line 234. N = 12? Please be explicit.

*Response: Thank you for the suggestion We added a supplementary explanation.*

Lines 240-241. To have "error (Max ERR) of the first decomposition component IMF1" you have to have "true" 2017 values for IMF1 as well as the predicted values. You need to say how you obtain the "true" values of IMF1. It seems to imply that you did the decomposition for 1982-2017 as well as for 1982-2016. Likewise for the other IMFi and RES. I guess the IMFs for these two decomposition periods differ in 1982-2016, although perhaps very little for IMF1 until the very end of 2016.

*Response: Thank you for the comment. We have actual values for 2017, so we can use it to compare against predicted values based on actual values for 1982-2016.*

Tables 1 and 2. The format of the values in any row should be the same for max/min/mean ERR and for RMSE.

*Response: Thank you for the comment. But We are sorry that we did not understand this and we thought we kept the same format -- all values reserved to the last four digits of the decimal point.*

Table 1 row RES. RMSE has to be greater than mean ERR. Also Mean ERR is too small; (Min ERR x 11 + Max ERR) / 12 exceeds Mean ERR.
*Response: Thank you for the comment. This is our typo error and we corrected it.*

Line 248. Delete "nonlinearity and" unless you can show non-linear dependence on an input.
*Response: Thank you for the suggestion We deleted it.*

Lines 273, 274. Please state a criterion for "satisfactory" and do not change it for CCEMD. At present you imply 0.3°C for EEMD and 0.1°C for CCEMD. And include the "°C".
*Response: Thank you for the comment. Indeed, how to evaluate satisfaction is a very difficult thing. However, we believe that the error of the prediction results obtained is less than 0.1°C, which is already a very good prediction result.*

Lines 281-283. Better "The correlation coefficient between the prediction values based on the CEEMD-BPNN model and observations is 0.97 indicating a significance level of 0.001. The result . . 2017 was predicted"
*Response: Thank you for the comment. We corrected it.*

Lines 286-287. Better ". . Table 3 shows that prediction results of the hybrid CEEMD and BPNN model are much better than with the EEMD-BPNN . ."
*Response: Thank you for the comment. We corrected it.*

Line 288. Delete "and nonlinear" unless you can show non-linear dependence on an input.
*Response: Thank you for the comment. We corrected it.*

Line 303. You can omit "respectively" (unnecessary).
*Response: Thank you for the comment. We corrected it.*

Line 307. ". . components (IMF1, . ."
*Response: Thank you for the comment. We corrected it.*

Line 308. Justify or omit "strong nonlinearity and non-stationarity. As the nonlinearity gradually decreases"
*Response: Thank you for the comment. We deleted it.*

Line 310. Better ". . preliminary, based . ."
*Response: Thank you for the comment. We corrected it.*

[revised manuscript text omitted]

---

## Author Response (AR3)

**Response to the comments**

*The topic editor knows the topic very well and his comments are indeed helpful in improving the quality of this MS. We are grateful to Prof. John M. Huthnance for a careful checking and comments on the MS. All comments are addressed point by point, each starting with an original comment and followed by a response in italic, as follows.*

Topic Editor Decision: Publish subject to minor revisions (review by editor) (08 Mar 2019) by John M. Huthnance

Comments to the Author:

Dear Authors

Thank-you for revisions. I am sorry that I still think there should be some improvements and I would like to see your manuscript once more.

*Response: Thank you for these comments. On behalf of my co-authors, we thank you for giving us an opportunity to improve our manuscript, we appreciate you very much for your constructive comments and valuable suggestions on our manuscript.*

*All of the following suggestions were accepted and we have made corresponding corrections in the revised manuscript.*

Line 25. Better ". . data of the individual IMFs. A case study . ."

Line 38. Better ". . 2019b) and dynamic processes"

Line 40. I do not understand "reflects". Maybe "e.g." or "and"?

Line 47. ". . randomness and irregularity of the . ."

Line 47 "high randomness" and line 55 "deterministic" are contradictory. You have to delete one of these.

Line 74. "component" – do you mean "IMF"?

Lines 76-77. These are about EEMD. Merge them in at lines 69-70.

Lines 86-87. This sentence is about EEMD. Merge it in at lines 69-70.

Then lines 78-85 and 87-91 about CCEMD are brought together but you will need to remove duplication.

Line 85. Do you mean ". . can effectively confine the impact of the time-series non-stationarity to the trending component and ??, which helps . .". If you can recombine the components to the original time series, then all the non-stationarity must remain in the components somewhere – where (replace the ?? above)?

Line 119. You need to define SSTA – i.e. anomaly relative to what?

Line 181. Better "reduced compared with that of the EEMD algorithm."

Lines 221-222. "observation results of 12 months in 2017 are used to compare and analyze with the prediction results." But you need to say how you go from the 2017 observations to the IMF1, IMF2, . . IMF7 for 2017 with which you compare the predicted IMF1, IMF2, . . IMF7.

Line 273. ". . less than 0.3°C). The . ." You must be consistent about what is "satisfactory". If ERR > 0.1°C is not satisfactory for CEEMD, as you imply, then 0.1°C < ERR < 0.3°C is not satisfactory for EEMD. I suggest for line 274 ". . satisfactory; ERR exceeds 0.1 °C only in October, . ."

[revised manuscript text omitted]